# aRDG Analysis of Asphaltene Molecular Viscosity and Molecular Interaction Based on Non-Equilibrium Molecular Dynamics Simulation

**DOI:** 10.3390/ma15248771

**Published:** 2022-12-08

**Authors:** Qunchao Lin, Lei Deng, Ge Dong, Xianqiong Tang, Wei Li, Zhengwu Long, Fu Xu

**Affiliations:** 1College of Civil Engineering, Xiangtan University, Xiangtan 411105, China; 2College of Aerospace Science and Technology, National University of Defense Technology, Changsha 410073, China; 3Science and Technology on Aerospace Chemical Power Laboratory, Hubei Institute of Aerospace Chemotechnology, Xiangyang 441003, China

**Keywords:** asphaltene, shear viscosity, NEMD, aggregation, weak interaction, aRDG

## Abstract

Understanding the noncovalent (weak) interactions between asphaltene molecules is crucial to further comprehending the viscosity and aggregation behavior of asphaltenes. In the past, intermolecular interactions were characterized indirectly by calculating the radial distribution function and the numerical distribution of distances/angles between atoms, which are far less intuitive than the average reduced density gradient (aRDG) method. This study selected three representative asphaltene molecules (AsphalteneO, AsphalteneT, and AsphalteneY) to investigate the relationship between viscosity and weak intermolecular interactions. Firstly, a non-equilibrium molecular dynamics (NEMD) simulation was employed to calculate the shear viscosities of these molecules and analyze their aggregation behaviors. In addition, the types of weak intermolecular interactions of asphaltene were visualized by the aRDG method. Finally, the stability of the weak intermolecular interactions was analyzed by the thermal fluctuation index (TFI). The results indicate that AsphalteneY has the highest viscosity. The aggregation behavior of AsphalteneO is mainly face–face stacking, while AsphalteneT and AsphalteneY associate mainly via offset stacking and T-shaped stacking. According to the aRDG analysis, the weak interactions between AshalteneT molecules are similar to those between AshalteneO molecules, mainly due to van der Waals interactions and steric hindrance effects. At the same time, there is a strong attraction between AsphalteneY molecules. Additionally, the results of the TFI analysis show that the weak intermolecular interactions of the three types of asphaltene molecules are relatively stable and not significantly affected by thermal motion. Our results provide a new method for better understanding asphaltene molecules’ viscosity and aggregation behavior.

## 1. Introduction

Asphaltene, the heaviest, densest, and most polar component in petroleum, is classified as a complex mixture of heavy organic molecules that are insoluble in n-alkanes, including n-heptane and n-pentane and soluble in aromatic solvents, such as toluene and benzene [1,2,3]. The viscosity of heavy oil is closely related to its asphaltene content, and the high concentration of asphaltene is the main reason for the high viscosity of heavy oil [4,5]. In addition, the aggregation behavior of asphaltene molecules is responsible for the deposition, emulsification, and high viscosity of heavy oil, which significantly affect its utilization and value.

As one of the important components of asphalt, asphaltenes are complex mixtures of polycyclic aromatic hydrocarbons substituted with alkyl side chains and heteroatoms, such as nitrogen, sulfur, and oxygen, which are typical substituents in conjugated cores [6,7]. In general, asphaltene molecules have a polycyclic aromatic core that is composed of approximately 4–10 aromatic rings and fatty side chains with a length of 3–7 carbon atoms [8,9]. In the past decade, various research groups have proposed and studied different asphaltene molecular models to study the mechanism behind asphaltene properties. Sjöblom [10] reviewed the different types of proposed asphaltene models and summarized the composition and properties of asphaltenes and the research methods and results for different types of model asphaltene compounds.

During the past few decades, people have been exploring the mechanism of asphaltene aggregation. Asphaltene molecules undergo aggregation for a number of reasons [11], including the extent of the conjugated core [6,12], the presence of heteroatoms [6,13], the length and polarity of the side chains [14,15], the type of solvent [16], temperature, and pressure [15], etc. In 1990, Hunter and Saunders [17] summarized three rules for porphyrins: (1) π-π repulsion dominates the face–face geometry; (2) π-σ attraction dominates the side geometry; and (3) σ-σ attraction dominates the offset π-stacking geometry. Pacheco [18] employed classical molecular dynamics (MD) to simulate asphaltene aggregation under vacuum at different temperatures and concluded that the aggregation behavior of asphaltene molecular dimers could also follow these rules.

Takanohashi [19] used MD simulation to study the stability of asphaltene aggregates of three model molecules at high temperatures and observed that aliphatic side chains and heteroatomic functional groups contribute to the stability of trimers. Rogel [20] used the average structure model to study the interaction force of the binding process between asphaltene and resin. The results indicate that the stabilization energy that was obtained by asphaltene and resin is due mainly to the van der Waals forces between the molecules. In contrast, the contribution of hydrogen bonds is low. Long et al. [21,22] investigated the effect of sea salt on the adhesion properties of asphaltene-agglomerate systems by molecular dynamics simulations. The results showed that asphaltenes are highly self-aggregating in dry environments, while water and seawater weakened the aggregation of asphaltene molecules. Seawater affects the adhesion by promoting the molecular migration of asphaltic components and weakening the asphaltene agglomeration. In addition, some researchers have studied the relationship between the asphaltene molecular structure and aggregation behavior [1,23,24].

Due to its many sources, asphaltene has a complex and diverse molecular structure, which has caused controversy for many years [4,21]. Schuler [25] utilized atomic force microscopy (AFM) and scanning tunneling microscopy (STM) to perform molecular orbital imaging for more than 100 asphaltene molecules and proposed that the molecular structure of asphaltene is an aromatic core, substituted with heteroatoms and a variable number of alkyl side chains. Long et al. [26] investigated the evolution of the microstructure of asphalt under chloride salt erosion by AFM, showing that chloride salt erosion caused the migration and aggregation of asphalt components (including asphaltenes), and that the formation of honeycomb structures on the asphalt surface was the result of the combined action of asphaltenes and waxes.

Sedghi [1] employed MD simulations to investigate the relationship between the aggregation and molecular structure of asphaltene. The results indicate that the interaction between the aromatic nuclei of asphaltene molecules is the main driving force of asphaltene aggregation. Furthermore, Jian [24] carried out a series of MD simulations to study the effect of aliphatic side chain length on the aggregation behavior of asphaltenes. The degree of aggregation is not monotonically related to the side chain length. Asphaltene molecules with short or long side chains can form dense aggregates, while those with medium-length side chains cannot. Ekramipooy [13] employed density functional theory and MD simulations to study the effect of heteroatoms on the aggregation of model asphaltenes. The results show that heteroatoms in the fatty side chain more effectively increase the aggregation. The heteroatoms in the middle of the fat side chain strengthen the CH…C dispersion interaction through carbon polarization.

Although the studies on asphaltene viscosity and asphaltene aggregation behavior alone have been numerous and intensive, there are few studies on the relationship between the two, which is an important research direction that is needed to understand the microscopic mechanism of asphaltene molecules with different viscosities, so this is one of the focuses of this study. In addition, most of these studies [13,20] perform indirect characterization by calculating radial distribution functions and the numerical distribution of interatomic distances/angles to study asphaltene interactions. Yang [27] proposed a visual method to study weak interactions as a direct characterization tool, referred to as the reduced density gradient (RDG) or noncovalent interaction (NCI) method. The so-called weak interaction refers to various forms of interactions whose strength is weaker than covalent chemical bonds, such as van der Waals interactions, π-π stacking, hydrogen bonds, halogen bonds, and dihydrogen bonds. To further identify the characteristics of the interactions between asphaltene molecules in a dimer, Wang [2] carried out NCI visualization analysis, focusing on the intermolecular interactions in the asphaltene dimer, and screened the intramolecular interaction of asphaltene, concluding that π-π interactions are the main driving force of asphaltene aggregation. Ekramipooya [13] adopted the RDG method to study the effect of heteroatoms on self-aggregation at different positions in a model of the asphaltene structure. By comparing the relationship between the RDG and sign (λ_2_) ρ of different asphaltene dimer models, Ekramipooya concluded that the heteroatoms at the X3 position (in the middle of the fat side chain) strengthened the CH…C dispersion interaction through carbon polarization. Yang [28] proposed the averaged RDG (aRDG) method (also known as the aNCI method), which is an extension of the original RDG method that can be used to analyze multi-frame structures, especially in combination with MD simulation techniques, to study weak interactions in equilibrium dynamic environments. Wu [29] investigated the aggregation behavior of three reactive dyes in water by MD simulations. The dye–water molecular interactions were analyzed by the RDG and aRDG methods, and the position, intensity, and type of the interactions were visualized. The results show that the main interactions between the dye and anions are van der Waals and π-π stacking interactions.

The aRDG method can be employed to study average weak interactions between small molecules in kinetic processes and receptor–protein, small peptide–small peptide, and molecule–solid surface cases [30,31,32,33]. At present, although MD simulations have been widely employed to investigate the molecular structure and aggregation behavior of asphaltenes, few studies have used the RDG method, especially the aRDG method [11], to examine the weak interactions between asphaltene molecules. In addition, non-equilibrium molecular dynamics (NEMD) has been used to study the shear viscosity of fluids by various techniques [22,34,35]. This method was used for the computational analysis of asphaltene molecular viscosity in this study.

Understanding the weak interactions between asphaltene molecules is essential for further understanding of asphaltene viscosity and aggregation behavior, and this study provides a novel approach to studying weak intermolecular interactions. In this study, three representative structures of asphaltene molecules were selected and optimized appropriately. MD simulations obtained the spatial aggregation morphology of three asphaltene molecules. Their shear viscosity was calculated by (NEMD) simulations to study the relationships among asphaltene molecular structure, viscosity, and aggregation behavior. Finally, the weak intermolecular interactions and their stability in the process of asphaltene aggregation were visualized by aRDG and TFI methods. Our results contribute to a better understanding of the variability in the viscosity and aggregation behavior of asphaltene molecules.

## 2. Simulation Methods

### 2.1. Molecular Model

In this work, the simulation study of model asphaltene was based on the structure of actual asphaltene molecules characterized by Schuler [36] using AFM and STM in 2017. We selected three types of asphaltene molecules with different arrangements and appropriate optimization, referred to as AsphalteneO, AsphalteneT, and AsphalteneY. The molecular structures are shown in Figure 1a. The cluster-like benzene ring structures are formed by several benzene rings, and the chain-like structures are formed by stacks of several benzene rings. AsphalteneO has a cluster-like aromatic ring structure, AsphalteneY has a branched chain-like aromatic ring structure, and AsphalteneT has both cluster-like and branched chain-like benzene ring structures [37,38]. It is worth mentioning that the focus of this study is to study the weak intermolecular interactions using the aRDG method. These few simple models are only used as a preliminary exploration to demonstrate the feasibility of the aRDG method for asphaltene studies and to provide technical and methodological support for future studies of asphaltene molecules with more complex structures.

In addition, the initial boxes of the three asphaltene molecules were established by Material Studio, as shown in Figure 1b. Each box contained a random distribution of 30 asphaltene molecules with an initial density of 0.6 g/cm^3^ for later NEMD simulation analysis.

### 2.2. NEMD Method

Non-equilibrium molecular dynamics (NEMD) has been employed to study the shear viscosity of fluids through various techniques [22,34]. The NEMD calculation imposes shear boundary conditions on the system of interest and is expected to drive it to a steady state. The dynamic behavior of the interatomic forces in the NEMD simulation was calculated by using the isothermal atomic formula of the SLLOD equation [39,40,41]. The velocity Verlet algorithm was used to solve the numerical integration of the velocities and positions of the particles [42]. In the NEMD calculation, the constant shear rate γ was applied to the infinite periodic array of the subsystem of volume V, and each subsystem contained N particles [34]. The shear viscosity η was calculated using the following formula:(1)$$\eta =–\frac{P_{\alpha \beta }}{\gamma }$$
where η is the viscosity, γ is the shear rate, and $P_{\alpha \beta }\left(\alpha ,\beta =x,y\right)$ is the pressure component of the system in Cartesian coordinates in the shear field.
(2)$$P_{\alpha \beta }=\frac{1}{V}\left(\sum_{i}m_{i}v_{i}^{\alpha }v_{i}^{\beta }+\sum_{i}\sum_{j>i}r_{ij}^{\alpha }F_{ij}^{\beta }\right)$$
where $m_{i}$ is the mass of the *i*th particle, $\nu_{i}$ is the relative velocity of the *i*th particle, $r_{ij}$ is the distance between particles i and j, and $F_{ij}$ is the force between particles i and j.

Material Studio software 2017 R2 was only used for modeling, and all of the MD simulations were performed with the large-scale atomic/molecular massively parallel simulator (LAMMPS) program package. The specific steps were as follows, as shown in Figure 2:

A. A box with 30 randomly distributed asphaltene molecules was created by Material Studio, and the initial density of the system was set to 0.6 g/cm^3^. This study used the COMPASS force field.

B. The density and energy of the randomly generated simulation box (including the kinetic and potential energy evaluation) were calculated. Each system was balanced by an NPT ensemble simulation under the constant pressure of 1 atm for 10 ns, and the temperature was kept at 333 K, giving the system a suitable density and box size.

C. The NVT simulation was carried out for 50 ns at 333 K using the Nose–Hoover thermostat. The shear rate was 1 × 10^−7^/fs, the time step was 1 fs, and the periodic boundary conditions were applied in all directions (under a high strain rate, the sharp increase in pressure prevented the NEMD method from calculating asphaltene viscosity [43], so the shear rate selected in this study was 1 × 10^−7^/fs.)

D. After 10 ns, the atomic displacement, force, velocity, potential energy, kinetic energy, viscosity, and other parameters that were related to the complete relaxation and equilibration of the system were recorded for subsequent analysis.

### 2.3. Averaged Reduced Density Gradient

#### 2.3.1. Reduced Density Gradient (RDG)

To reveal the weak interactions, Johnson et al. [27] constructed the formula based on the study of reduced density gradient (RDG) as a function of electron density  ρ(r):(3)$$\mathrm{RDG}\left(r\right)=\frac{1}{2{\left(3\pi^{2}\right)}^{1/3}}\frac{\left|\nabla \rho \left(r\right)\right|}{\rho {\left(r\right)}^{4/3}}$$
where ∇ is the gradient operator, ρ(r) is the electron density, and  |∇ρ(r) | is the norm of the electron density gradient.

RDG, a method to characterize and visualize different types of weak interactions, defines an actual space function that enables its value to distinguish between regions with different characteristics in the system. Suppose a particular weak interaction has a larger ρ(r) at its critical point. In that case, ρ(r) will generally be larger in the surrounding region due to the continuity of ρ(r). Therefore, the strength of the interaction is apparent at a glance by mapping the numerical magnitude of ρ(r) to the RDG equivalent surface in different colors. However, ρ(r) can only reflect the strength of the interaction, and the type of interaction needs to be reflected by the sign (λ_2_) function (where λ_2_ is the second largest eigenvalue of the Hessian matrix of electron density). The electron density ρ(r) and the sign (λ_2_) function are combined to obtain the sign (λ_2_) ρ function that is projected onto the RDG equivalent surface, which reveals the location, strength, and type of the weak interaction. The color scale is set to blue, green, and red. The blue region exhibits more potent, attracting weak interactions, and the green region has a small ρ(r), indicating weak interaction strength. The van der Waals interaction region fits this feature. The red region corresponds to the stronger region of the potential barrier effect (also known as nonbonded overlap) in the benzene ring. Specific applications and analyses are discussed in detail in Section 3.

#### 2.3.2. Averaged Reduced Density Gradient (RDG)

The aRDG method is very similar to the RDG method in principle. Similar to the RDG method, the aRDG method shows the weak interaction region through the equivalent surface of the RDG function. It projects the sign (λ_2_) ρ function on the RDG equivalent surface with different colors to show the weak interaction types. However, in the aRDG method, the electron density, the gradient of the electron density, and the Hessian matrix of the electron density are used to calculate the RDG and sign (λ_2_) ρ functions, which are all obtained by averaging over the multi-frame structure. Therefore, both the RDG function and the sign (λ_2_) ρ function in the aRDG method represents the average RDG and the average sign (λ_2_) ρ for the whole trajectory. The calculation formulas are as follows:(4)$$a\mathrm{RDG}\left(r\right)=\frac{1}{2{\left(3\pi^{2}\right)}^{1/3}}\frac{\left|\bar{\nabla \rho \left(r\right)}\right|}{{\bar{\rho \left(r\right)}}^{4/3}}$$
where ▽ is the gradient operator, ρ(r) is the electron density, |▽ρ(r) | is the norm of the electron density gradient, $\bar{\rho \left(r\right)}$ is the average electron density, and $\left|\bar{▽\rho \left(r\right)}\;\right|$ is the norm of the average electron density gradient. In addition to this difference, the aRDG method defines a thermal fluctuation index (TFI) that can be calculated using the following formulas:(5)$$\mathrm{TFI}\left(r\right)=\frac{std\left[\rho \left(r\right)\right]}{\bar{\rho \left(r\right)}}$$
(6)$$std\left[\rho \left(r\right)\right]=\sqrt{\frac{\sum_{i}{\left[\rho_{i}\left(r\right)-\bar{\rho \left(r\right)}\right]}^{2}}{n}}$$
where $\bar{\rho \left(r\right)}$ is the average density in the whole trajectory mentioned above, and std[ρ(r)] is the standard deviation of the electron density in the dynamic locus. The greater the standard deviation of a function, the stronger the fluctuation of the function. The TFI shows the stability of the weak interaction in different regions. The TFI was mapped to the average RDG equivalent surface through different colors during the analysis. By examining the color, we could judge whether the weak interaction in different regions was stable.

The asphaltene molecular trajectory file was obtained in this study by NEMD simulation. Then, the asphaltene molecular interaction and its stability were analyzed by aRDG and TFI using Multiwfn 3.8 software [44] that was developed by the Lu Tian team. Since the difference between promolecular density and the actual density in the weak interaction region is not as significant as in the bonding region, promolecular density can be an approximate substitute for actual electron density in the aRDG method. In this process, promolecular density [28] built-in Multiwfn software was applied to reduce the calculation time. The results of the aRDG and TFI analysis in this study can be displayed by the VMD program [45].

## 3. Results and Discussion

### 3.1. Shear Viscosity

Figure 3 shows the viscosity changes of the three kinds of asphaltene molecules during the NEMD simulations. The average viscosity data were calculated every 10 steps, 50 steps, 100 steps, and 1000 steps. The viscosities obtained at the four frequencies converged utterly, and the convergence values were almost the same. The total time of the simulation process was 60 ns. Due to the excessive fluctuations in the data during the first 10 ns, statistical analysis was carried out using the data collected after 10 ns. Finally, the viscosity changes of three asphaltenes during 10–60 ns were obtained. The viscosity began to converge after 55 ns. The viscosities of the three asphaltenes, 313 cP (AsphalteneY), 274 cP (AsphalteneT), and 90 cP (AsphalteneO), were obtained by taking the average value between 55 and 60 ns, indicating that the viscosity of asphaltene molecules is closely related to the structure.

Asphaltene molecules with different structures have different aggregation mechanisms. Therefore, to further understand the viscosity of asphaltene, the aggregation behavior of asphaltene is analyzed in the following section, which provides conditions for explaining the differences in the molecular viscosities of the asphaltene molecules with different structures.

### 3.2. Asphaltene Aggregation

Hunter and Saunders [17] summarized three rules about porphyrins as shown in Figure 4. Pacheco [18] employed classical molecular dynamics (MD) to simulate asphaltene aggregation under a vacuum at different temperatures and concluded that the aggregation behavior of asphaltene molecular dimers could also follow these rules, as shown in Figure 5.

In this study, the three asphaltene molecules are classified as AsphalteneO, AsphalteneT, and AsphalteneY according to their structure. The three asphaltene molecules after NEMD simulation are visualized by VMD 1.9.4 software to observe the specific aggregation patterns of asphaltene molecules with different structures, as shown in Figure 6.

Most of the AsphalteneO molecules aggregated via face–face stacking, and a few molecules underwent face–face stacking and offset stacking. AsphalteneT has both cluster-like benzene ring and chain-like benzene ring structures; the cluster-like benzene ring structures of AsphalteneT molecules stack face–face, but the chain-like benzene ring structures are staggered, which is classified as offset stacking in this study. In addition, some AsphalteneT molecules form T-shaped aggregates where the cluster-like benzene ring structure of one molecule is in contact with that of another molecule, and the two molecules are distributed vertically with an overall T-shape. On the other hand, AsphalteneY mainly forms T-shaped or cross-shaped aggregates that are related to the molecular structure. Some molecules associate via offset stacking because they contain chain-like benzene ring structures.

According to the previous section, the viscosities of the three asphaltene molecules are in the order AsphalteneY > AsphalteneT > AsphalteneO, which can be fully explained by the aggregation behaviors in this section. AsphalteneO tends to undergo face–face stacking under the conjugation effect of cluster-like benzene ring structures. AsphalteneO molecules move relatively smoothly under the action of a shear field. However, in addition to offset stacking, AsphalteneY undergoes T-shaped cross stacking between the molecules, which appears to hinder the movement of the molecules structurally. AsphalteneT has the conjugation effect of cluster-like benzene ring structures and the structural hindrance effect of chain-like branched benzene ring structures. In other words, different aggregation methods lead to different viscosities. Asphaltene viscosity is a macroscopic expression of asphaltene molecule aggregation, and asphaltene molecules showing T-shaped stacking aggregation have higher viscosities (AsphalteneY and AsphalteneT), while asphaltene molecules showing face–face stacking aggregation have lower viscosities (AsphalteneO).

Moreover, we can guess that the interactions between AsphalteneY molecules should be larger and more complex than those of AsphalteneO, making their rupture recombination behavior under the shear field different. Therefore, we will next use aRDG analysis to reveal the intermolecular interactions of the three asphaltenes and provide data to support this conjecture.

### 3.3. aRDG Analysis

Figure 7 shows the plot of aRDG versus the average sign (λ_2_) ρ between the three asphaltene molecules. The spikes at different positions represent different types of weak interactions. For example, in Figure 7a, the spike near −0.03 represents attraction, the spike in the −0.18–0.16 interval represents the van der Waals interaction, and there are two distinct spikes in the 0.02–0.1 interval, representing two different degrees of intermolecular repulsion. The other four plots have the same pattern. However, sometimes the spikes are not as obvious as in Figure 7, and then further analysis is needed in conjunction with Figure 8.

The asphaltene molecules’ intermolecular interaction types and stabilities can be directly visualized by using aRDG and TFI analysis. In the aRDG analysis, we used the following colors to identify the types of interactions: blue indicates strong attractive interactions, such as hydrogen bonds; green represents weak interactions, such as van der Waals interactions; and red indicates repulsive interactions, such as steric hindrance, as shown in Figure 8k. In the TFI analysis, blue was used to describe interactions that were almost unaffected and highly stable when undergoing thermal motion, red indicated interactions that were easy to destroy and unstable during thermal motion, and green was used to indicate interactions that fluctuated between the “blue” and “red” types. The results of aRDG and TFI analyses are shown in Figure 8.

The large green area in Figure 8a shows a wide range of van der Waals interactions between AsphalteneO molecules, and the corresponding spike-tip x value on the scatter diagram is approximately ±0.015, as shown in Figure 7a. There is a red spindle region in the middle of the benzene ring, showing a strong steric effect, which corresponds to the spike on the far right of the scatter diagram. Notably, there is an area between C-H and C-H, half of which is red and half is blue, corresponding to spikes with x values of approximately +0.03 and −0.03, respectively. This isosurface shows both a steric effect and an attraction effect, and mutual exclusion and attraction effects coexist. However, there is no strong attraction between C-H and C-H. This part experiences only van der Waals interactions because aRDG is based on the excimer density approximation, so the electron density in some weak interaction regions is not sufficiently realistic and may be overly high. According to the analysis of the TFI in the figure ( blue area in Figure 8b ), the weak interaction between AsphalteneO molecules has little influence on thermal motion and is relatively stable in the system.

On the one hand, Figure 8a,c show that the weak interactions between AsphalteneT and AsphalteneO molecules are very similar. On the other hand, the van der Waals forces between the T-shaped stacked AsphalteneT molecules, such as the two small green areas shown in Figure 8e, are weaker. The spike corresponding to the scatter plot is thinner than that of the offset-stacked AsphalteneT molecules, as shown in Figure 7b,c.

Compared with the interactions of AsphalteneT and AsphalteneO, the weak intermolecular interactions of offset-stacked AsphalteneYs are slightly more complicated; AsphalteneY molecules do not have strong van der Waals interactions. Figure 8g shows that green and blue merge on the equivalent surface of AsphalteneY. For another stacking method of AsphalteneY, as shown in Figure 8i, the equivalent surfaces between the two molecules are green and blue. There are van der Waals interactions between the chain-like structures of the benzene rings. However, compared with the other two kinds of asphaltene molecules, AsphalteneY has more blue isosurfaces and stronger attractions, which is one of the most significant reasons it has the highest viscosity in this study. Meanwhile, the results of the analysis verify the conjecture in the previous section.

Additionally, according to the TFI analysis, the weak intermolecular interactions of the three kinds of asphaltenes are relatively stable and slightly affected by thermal motion, as shown in Figure 8b,d,f,h,j.

## 4. Conclusions

This study selected three representative asphaltene molecules (AsphalteneO, AsphalteneT, and AsphalteneY) to investigate the relationship between the weak intermolecular interaction of asphaltene molecules and viscosity using NEMD simulations and the aRDG method.

The results indicate that the viscosity of asphaltene molecules is likely to be related to their intermolecular interactions with the order of viscosity values being AsphalteneY > AsphalteneT > AsphalteneO. One of the possible reasons for this is that, compared to AsphalteneT and AsphalteneO, the weak AsphalteneY intermolecular interactions are slightly more complex and include stronger attractive forces. The weak interactions between AsphalteneT molecules are similar to those between AsphalteneO molecules, mainly van der Waals interactions and steric hindrance effects. On the other hand, different asphaltene molecular structures exhibit different aggregation patterns, which affect the motion of asphaltenes under the action of shear fields and thus exhibit different viscosities at the macroscopic level.

In addition, TFI analysis was used to visualize the weak intermolecular interactions and their stability for asphaltene molecules. The weak intermolecular interactions of the three asphaltene molecules have good stability and are slightly affected by thermal motion.

The asphaltene molecular models that were employed in this study are relatively simple (containing only aromatic rings). At the same time, alkyl chains and heteroatoms (O, S, N) are also essential factors that affect the viscosity, aggregation behavior, and other properties of asphaltenes. Therefore, in the future, we intend to investigate the possible effects of alkyl chain length and heteroatoms on the viscosity and aggregation behavior of asphaltenes. We plan to employ the aRDG method to study the weak intermolecular interactions of asphaltene molecules with different lengths of alkyl chains and with and without the addition of heteroatoms.

## Figures and Tables

**Figure 1 materials-15-08771-f001:**
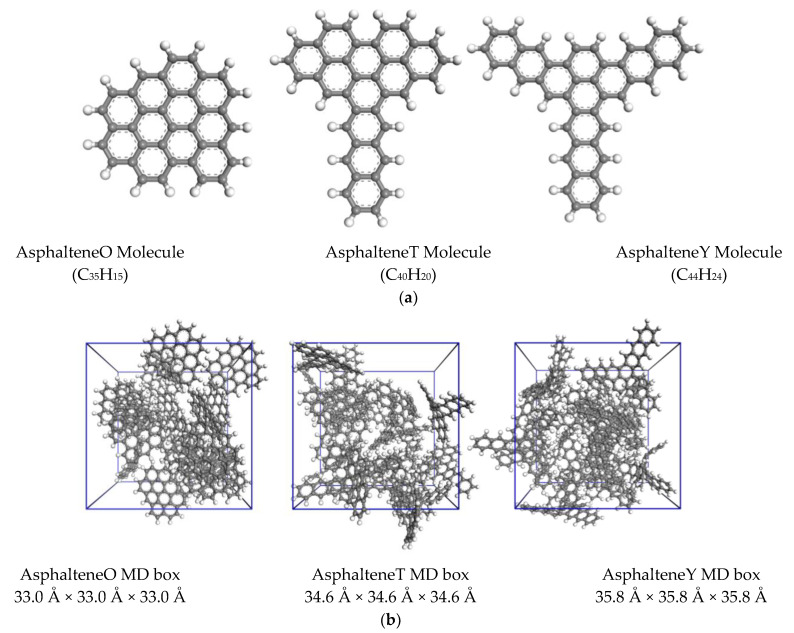
The molecular structure of the asphaltene molecules. Color coding of various elements: white, H; gray, C. (**a**) Molecular structures; (**b**) molecular dynamics modeling box.

**Figure 2 materials-15-08771-f002:**
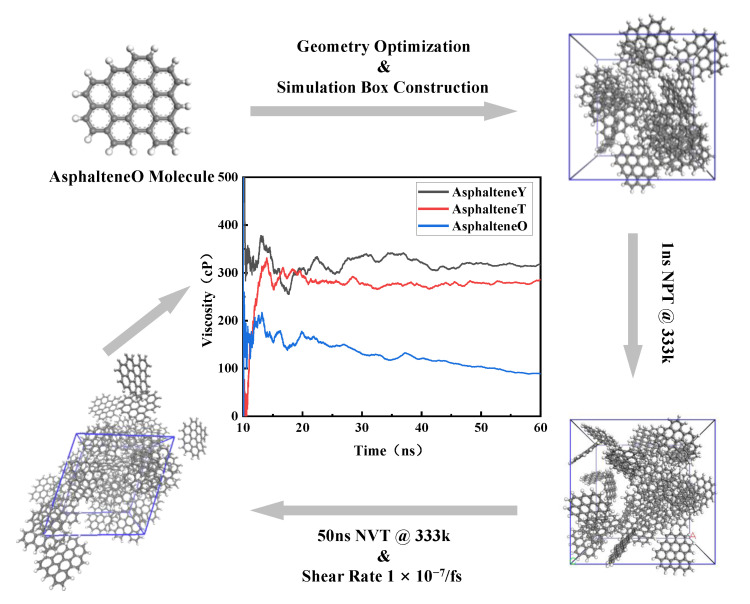
Main stages of calculating asphaltene molecular viscosity by NEMD simulation.

**Figure 3 materials-15-08771-f003:**
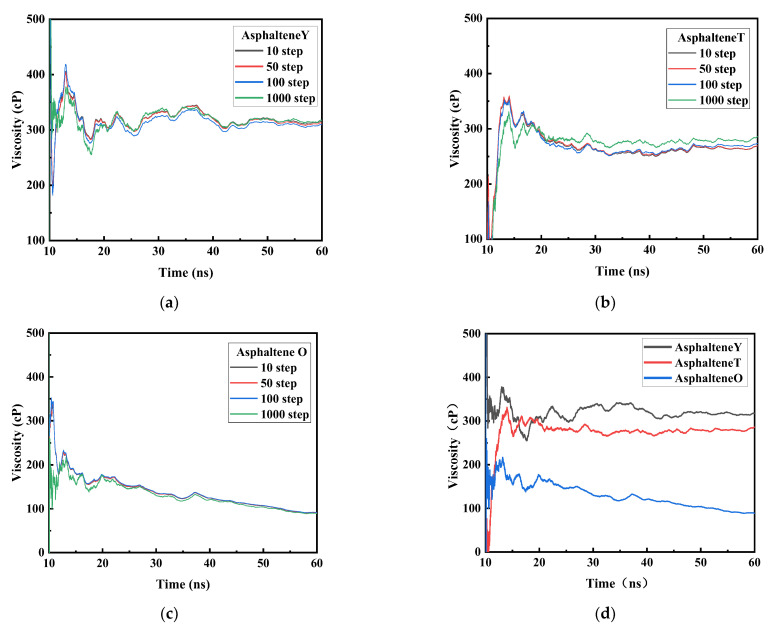
Viscosity change diagrams of three kinds of asphaltene molecules during the NEMD simulations process. (**a**) AsphalteneY, (**b**) AsphalteneT, (**c**) AsphalteneO, and (**d**) AsphalteneY, AsphalteneT and AsphalteneO.

**Figure 4 materials-15-08771-f004:**
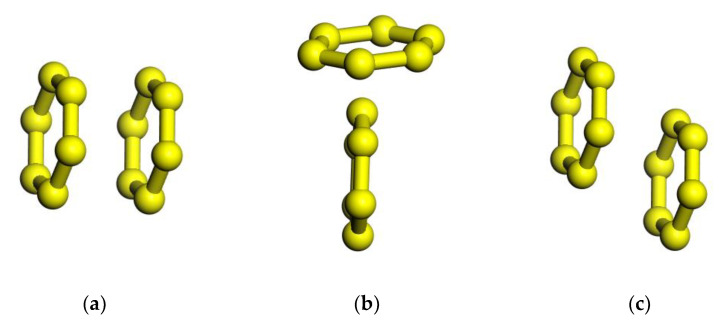
Three kinds of different stacking methods of aromatic rings. (**a**) Face–face stacking, (**b**) edge-on or T-shaped stacking, and (**c**) offset stacking.

**Figure 5 materials-15-08771-f005:**
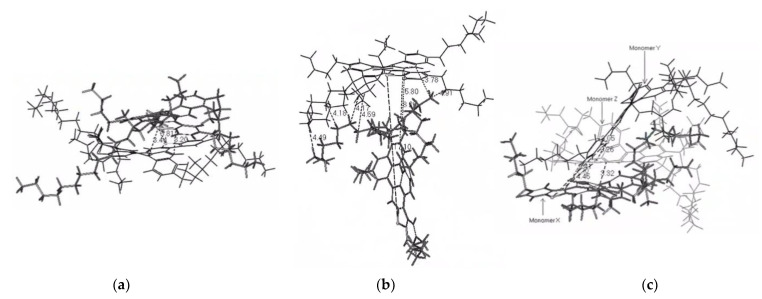
The aggregation behavior of asphaltene molecular dimers [18]. (**a**) Face–face geometry, (**b**) T-shaped or edge-on geometry, and (**c**) offset π-stacked geometry.

**Figure 6 materials-15-08771-f006:**
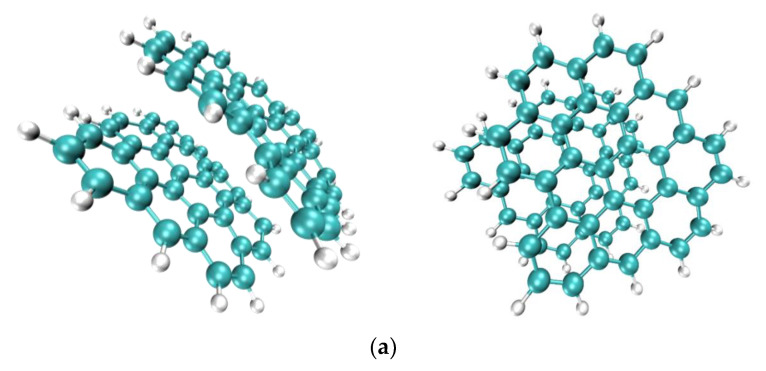
Aggregation behavior of three asphaltene molecules. (**a**) AsphalteneO: face–face stacking; (**b**) AsphalteneT: offset stacking (left) and T-shaped stacking (right), and (**c**) AsphalteneY: offset stacking (left) and crossed stacking (right).

**Figure 7 materials-15-08771-f007:**
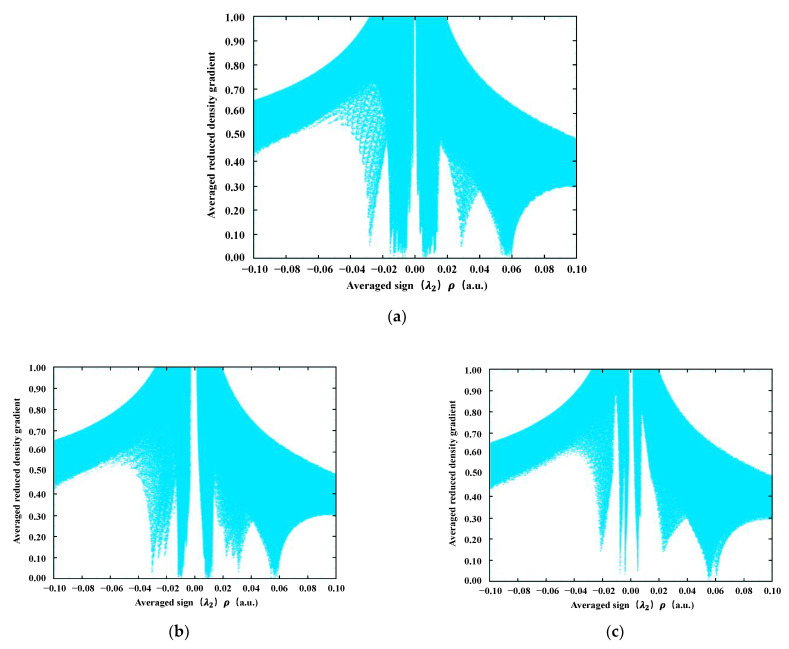
Diagrams of aRDG vs. averaged sign (λ_2_) ρ for three asphaltene molecules. RDG isosurfaces colored according to an RGB scheme over the range of 0.1 < sign (λ_2_) ρ < 0.1 a.u. (**a**) AsphalteneO, (**b**) AsphalteneT offset stacking, (**c**) AsphalteneT T-shaped stacking, (**d**) AsphalteneY offset stacking, and (**e**) AsphalteneY crossed stacking.

**Figure 8 materials-15-08771-f008:**
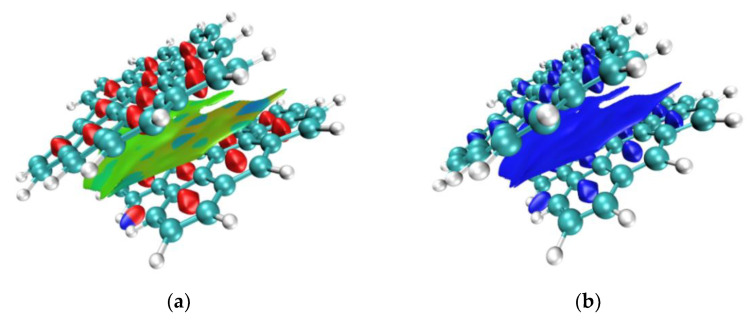
The results of aRDG (**a**,**c**,**e**,**g**,**i**) and TFI (**b**,**d**,**f**,**h**,**j**) analysis of three kinds of asphaltene molecules. The colors identify the types of interactions (**k**).

## Data Availability

Not applicable.

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
