# Peer review of "aRDG Analysis of Asphaltene Molecular Viscosity and Molecular Interaction Based on Non-Equilibrium Molecular Dynamics Simulation"

_materials, 2022, doi:10.3390/ma15248771_

Round 1

Reviewer 1 Report

This study selected three asphaltene molecules to investigate the relationship among the interaction of asphaltene molecules, aggregation behavior, and viscosity using NEMD simulations and the aRDG method. Overall, this is interesting research.  But there are still some problems needed to be explained:

1-    It is not usual to use the figure in the introduction. It is suggested to remove Figure 1 from the introduction and use it in another part of the manuscript.

2-     Parameter ρ(r) is shown as ρ(r) in line 209 and elsewhere in the manuscripts. Is ρ(r) the same as ρ(r)? If yes, please correct it.

3-    There is no explanation given in the text about figures 7(d) and 7(e).

4-    It is suggested that the necessity of conducting this research should be stated in the last paragraph of the introduction.

5-    The studies that have used the "NEMD Method" in the literature should be summarized in a paragraph in the introduction.

6-    What effect can the type of aggregate have on the results? Please provide a brief description of this in the results section.

7-    What is the relationship between asphaltene molecules (AsphalteneO, AsphalteneT, and AsphalteneY) and the properties of the asphalt binder, such as the degree of penetration, softening point, or the performance grade of the asphalt binder? Is there a relationship? It is important to explain how findings can be used pavement engineers.

8-    What kind of aggregate and what type of asphalt binder should be used for a better connection between them? Do the authors have any advice in this regard?

9-    The English writing of the presented manuscript need to be revised.

10- Please check the following article:

Bian, He, et al. "Insight into the mechanism of asphaltene disaggregation by alkylated treatment: An experimental and theoretical investigation." Journal of Molecular Liquids 343 (2021): 117576.

Reviewer 2 Report

This paper deals with the asphalten molecules interactions linked to the viscosity. The authors did a well done introduction of the subject including the domain and how molecular simulations were used in the field. The authors introduced the molecular models and the two kind of calculations they did: viscosity from non equilibrium molecular dynamics (NEMD) and non covalent interaction analyses from reduced density gradient (RDG) or averaged RDG (aRDG) methodologies. The aim of work and the idea to describe a link between viscosity and molecular interaction is a good point and a direction to follow. This could improve the understanding of the asphaltene fraction behavior associated to microscopic phenomenon. Nevertheless, there are several defaults in the pathway followed in this work and it cannot be published as this in particular because of the choice of the molecular models and the design of the simulation boxes which conduct to obvious conclusions.

Hereafter are some comments. I hope they could help to improve the work.

1) Introduction : in the introduction, speaking about asphalted aggregation models, although the authors give several references, they did not place their work relatively to the commonly accepted asphalten aggregation models of Gray (10.1021/ef200654p) and Mullins (10.1021/ef300185p).

2) Concerning the choice of the asphaltene models, in the conclusion the authors open the possibility of future works including chemical diversity. But I think this was crucial and a major failure of the work. All along the introduction (line 34, line 43, lien 81 ...) they indicate that asphaltene molecules include heteroatoms. This is why they are so polar and why they have a so complex aggregation behavior. They cite the work of Schuler (line 79) who did AFM experiments and showed that asphaltene contain heteroatoms. But in this work the chosen molecular model are apolar (figure 2). Several works showed that such models with only C and H do not behave as asphaltene and are soluble in heptane for example (Villegas 10.1021/acs.energyfuels.0c02744, Headen 10.1021/acs.energyfuels.8b03196, Santos-Silva 10.1021/acs.energyfuels.6b01170). Moreover, I disagree with the assertion that hydrogen bonds have a low effect (line 69). The same authors showed that in the case of the presence of heteroatoms, the role of hydrogen bonds is crucial to build large asphalten aggregates and lead to precipitation.

The sentence saying that aspaltene models are representative is wrong (line 135, line 418). The authors have to include representative models in their work and this will lead to a nice paper. This is effectively a huge amount of additional work but the methodology or strategy described and used in this paper is not enough new and cannot make by itself a paper.

3) Concerning the methodology. The authors only consider boxes with asphalten molecules, or they omitted to explain if they putted a solvent in the box. I could understand this choice in the case of the study of the viscosity. They could want to compute the viscosity of a bulk asphaltene fraction, previously to any chemical action or dissolution. But in the case of the analysis of the interaction and, in particular, when they compute the number of molecules per aggregate (figure 6) this is irrelevant. 

I do not understand the results of figure 6. If the box is only a box with asphalten molecules, and if you are in a condensed phase, you have only one uniq aggregate. How can you obtain 6 different aggregates (each with 5 molecules) ? It means they are not in contact ? But you are in a condensed phase with only asphalten.

The authors have to clearly explain how do they determine that two asphalten molecules belong to the same aggregate and precise the aim of this aggregation number in this context.

4) Concerning the RDG and aRDG calculations, the authors have to explain what is the electronic density they are using. Line 255 and 257 the authors said that they used MultiWFN to compute aRDG and pro-molecular density. What does this mean ? Such RDG analyses need the electronic density of the dimer and it is not clearly explained how the authors compute that density. They have to give the quantum chemistry framework they used to get and represent the electronic density.

5) Line 285, this is a semantic issue. VMD does not perform any visual analyses. The authors have to say how they selected the configurations of the dimers.

6) Finally, because of the choice of the molecular model. The conclusion about the fact that there is only weak interactions is obvious (line432). As only non polar molecules are considered, interactions are weak. Including heteroatoms will enhance the polarity of the molecules and will lead to a larger heterogeneity in the interactions.

The interaction between the aromatic nucleus (T-shape, offset and pi-stacking) are well know situations. The fact the authors obtained such conformation is not a result. By the way, they still could use this vocabulary to classify particular configurations.

Reviewer 3 Report

This is a very careful and well-written work on non-equilibrium molecular dynamics investigating the aggregation properties of polyaromatic molecules. The manuscript presents a thorough overview of methods and results, with clear conclusions.

I suggest publication in its present form.

Reviewer 4 Report

The paper "aRDG analysis of asphaltene molecular viscosity and aggregation behaviors based on NEMD simulation" presents a relevant theme and within the scope of this journal, and can be considered after some corrections suggested below:

(a) It is not recommended to use acronyms in the title, such as "NEMD", this should also be checked in the abstract. Check the wording of the entire title;

(b) The abstract is generally well written, however in terms of content it is generic, i.e., the authors lack an in-depth study of the quantitative results of this research;

(c) Scientific innovation is limited in the introduction of the paper, the authors must go deeper and detail what this research differs from countless others that exist on this topic, this must be evidenced together with the objectives at the end of the introduction;

(d) The state of the art is limited, note that in the introduction and throughout the paper there are few references, in addition the citation standard is different from that used by MDPI;

(e) Combined Figures must have their captions reformulated, note that the presentation mode differs from that used in the rules for journal authors;

(f) Discussions should be based in more depth, especially with comparisons with other studies in the literature on the subject studied;

(g) The conclusion is not very objective, the authors can reformulate their approach and way of presenting the conclusions to the authors.

The paper "aRDG analysis of asphaltene molecular viscosity and aggregation behaviors based on NEMD simulation" presents a relevant theme and within the scope of this journal, and can be considered after some corrections suggested below:

(a) It is not recommended to use acronyms in the title, such as "NEMD", this should also be checked in the abstract. Check the wording of the entire title;

(b) The abstract is generally well written, however in terms of content it is generic, i.e., the authors lack an in-depth study of the quantitative results of this research;

(c) Scientific innovation is limited in the introduction of the paper, the authors must go deeper and detail what this research differs from countless others that exist on this topic, this must be evidenced together with the objectives at the end of the introduction;

(d) The state of the art is limited, note that in the introduction and throughout the paper there are few references, in addition the citation standard is different from that used by MDPI;

(e) Combined Figures must have their captions reformulated, note that the presentation mode differs from that used in the rules for journal authors;

(f) Discussions should be based in more depth, especially with comparisons with other studies in the literature on the subject studied;

(g) The conclusion is not very objective, the authors can reformulate their approach and way of presenting the conclusions to the authors.

Round 2

Reviewer 1 Report

The comments have been considred. 

Reviewer 2 Report

Dear Editor

Authors replied to all the questions pointed out in the previous report. Here are minor revisions to be included before publication.

1) Author said in their report:

However, the author believes that the focus of this study is to study the weak intermolecular interactions using the aRDG method. This simple model is only used as a preliminary exploration to demonstrate the feasibility of the aRDG method for asphaltene studies and to provide technical and methodological support for future studies of asphaltene molecules with more complex structures.

I think this comment should be added in the text when the molecular models are introduced (section 2.1) or at the end of the introduction. This is crucial to higlight that the aim of the paper is to bring a proof of concept of the use of aRDG.

2) Concernant promolecular density, in addition to the reference. Authors may add a sentence explaining that the approximation of considering only the sum of free atom density is relevant in particular for these large systems. For example, line 260 or around.

3) In this same place, authors should mention on how much frames the average is computed. In particular because of the non-equilibrium character of the simulations, in is important to know which part of the trajectory is considered. 

4) Line 201: The temperature of 333K leads to a given density that seems to be "correct". Precise what does correct means here ? Is the density compared to experimental value ?

5) Lines 353-354, there are typos about the intervals.

6) Finally, in the title or in the conclusion, line 429, the mention "aggregation behavior" should be changed to molecular interaction. In the work done in this papers, authors consider a pure phase of asphalten molecules, again, as in my previous report, I think that speaking about aggregation is not relevant.  The results highlight how molecules are in interactions.
